# Heat Shock Proteins and Ovarian Cancer: Important Roles and Therapeutic Opportunities

**DOI:** 10.3390/cancers11091389

**Published:** 2019-09-18

**Authors:** Abdullah Hoter, Hassan Y. Naim

**Affiliations:** 1Department of Biochemistry and Chemistry of Nutrition, Faculty of Veterinary Medicine, Cairo University, Giza 12211, Egypt; abdullah.hoter@vet.cu.edu.eg; 2Department of Physiological Chemistry, University of Veterinary Medicine Hannover, 30559 Hannover, Germany

**Keywords:** ovarian cancer, heat shock proteins (HSPs), clusterin, therapeutic resistance, HSP inhibitors, ovarian cancer treatment

## Abstract

Ovarian cancer is a serious cause of death in gynecological oncology. Delayed diagnosis and poor survival rates associated with late stages of the disease are major obstacles against treatment efforts. Heat shock proteins (HSPs) are stress responsive molecules known to be crucial in many cancer types including ovarian cancer. Clusterin (CLU), a unique chaperone protein with analogous oncogenic criteria to HSPs, has also been proven to confer resistance to anti-cancer drugs. Indeed, these chaperone molecules have been implicated in diagnosis, prognosis, metastasis and aggressiveness of various cancers. However, relative to other cancers, there is limited body of knowledge about the molecular roles of these chaperones in ovarian cancer. In the current review, we shed light on the diverse roles of HSPs as well as related chaperone proteins like CLU in the pathogenesis of ovarian cancer and elucidate their potential as effective drug targets.

## 1. Introduction

### 1.1. Ovarian Cancer Is a Serious Problem in Gynaecological Oncology 

Ovarian cancer (OC) is a major life-threatening problem in the field of gynecological oncology. Globally, it stands as the foremost cause of death in women accounting for approximately 239,000 newly diagnosed cases and over 150,000 deaths per year [1]. Recent reports in the United States estimated 22,240 new cases with ovarian cancer and 14,070 deaths owing to the disease [2]. Notably, the highest incidence and mortality rates have been linked to Eastern and Central Europe [1]. Therefore, great efforts are required to improve the therapeutic outcomes for diseased women. Additionally, thorough understanding of the molecular mechanisms and key elements contributing the disease is substantial in combating ovarian cancer [3]. 

Indeed, ovarian tumors can arise from three ovarian cell types namely, surface epithelium, sex cord stromal cells and germ cells [4]. Epithelial tumors account for 90% of ovarian malignancies while non-epithelial tumors including sex cord stromal and germ cell tumors represent 10% of the diagnosed cases. Epithelial ovarian cancer (EOC) are histologically categorized into serous, endometrioid, clear cell and mucinous carcinomas; the serous type itself is subclassified into high grade serous carcinoma (HGSC), low grade serous carcinoma (LGSC) and serous tubal intraepithelial carcinoma (STIC) [3] (a brief classification of OC histology is illustrated in Figure 1).

OC is often diagnosed at relatively old age of life, with a median age of 63 years in the US women population (https://seer.cancer.gov/statfacts/html/ovary.html). In addition, current data show that 59% of the cases have metastatic forms of the disease, while only 15% are diagnosed at the local stage. Of particular importance, early detection of ovarian malignancies is associated with higher cure rates, with a five-year survival exceeding 92% for localized ovarian cancer, whereas late stage diagnosis of the metastatic disease lowers cure rates to 20% [5,6].

The standard treatment protocol for human ovarian cancer includes maximal cytoreductive surgical debulking followed by the platinum-based chemotherapy. Concurrent with surgical cytoreduction, staging of the disease remains important [7,8]. Current therapeutic regimens to the first-line treatment which involve bevacizumab and paclitaxel have shown improved survival among patients with OC [7,9]. Unfortunately, despite initial remarkable response to chemotherapy, the majority of advanced OC cases recur after primary drug treatment with fatal outcome [10]. According to Ovarian Cancer Research Alliance (OCRA), current reports show that patients diagnosed at stages I and II have a recurrence chance of 10% and 30%, respectively, whereas the chance of recurrence in those of stage III and IV ranges between 70% and 95% (https://ocrahope.org/patients/about-ovarian-cancer/recurrence/). 

Multiple treatment approaches have been adapted for management of relapsed ovarian cancer. For instance, agents targeting angiogenesis include Bevacizumab, a monoclonal antibody that binds human vascular endothelial growth factor (VEGF) and inhibits its activity. Cediranib is an oral VEGF receptor and c-KIT inhibitor that displays antitumor activity in relapsed EOC in phase I/II studies. Trebananib is a peptide that suppresses angiogenesis by inhibiting angiopoietin-1 and -2. Moreover, other treatment strategies involve PARP inhibitors (PARPi) which render PARP enzymes no more capable of performing DNA repair processes and ultimately leading to synthetic lethality [11]. These PARP inhibitors include olaparib (AZD2281), niraparib (MK4827), rucaparib (CO338, AGO14699, and PF01367338), veliparib (ABT-888) and talazoparib (BMN 673) [11]. However, it should be noted that that PARP inhibitors have mostly been successful and are approved for patients with platinum sensitive ovarian carcinoma rather than resistant disease [12]. Furthermore, recent reports show that sorafenib, a pleiotropic tyrosine kinase inhibitor that inhibits pathways mediated by angiogenic and growth stimulating factors, could significantly increase the progression-free survival in platinum-resistant OC patients compared to placebo [13].

A common significant problem in the treatment of women with advanced OC is therapeutic resistance. To enhance therapeutic outcomes in recurrent OC, it would be beneficial to generate drugs or therapeutic combinations that would overwhelm the resistance and increase the response to the main therapy. Therefore, targeting the molecular mechanisms underlying such drug resistance in OC is highly recommended and may allow for optimum treatment [14].

Recent research highlights the implication of heat shock proteins (HSPs) in malignant processes and their association with drug resistance in cancer. HSPs are known as stress inducible molecules that are highly conserved across prokaryotic and eukaryotic species ranging from bacteria to human [15,16]. These molecules are well known for their molecular chaperone activities including protein folding, anti-aggregation of proteins and cellular protein trafficking [17,18,19,20]. Expression of HSPs is either constitutive or induced by various physiological, environmental and pathological factors including thermal stress, hypoxia, inflammation, toxic agents, heavy metals and cancer [21]. In response to variant stresses, members of HSPs are mostly regulated by a physiological process collectively known as “heat shock response (HSR)”, which involves heat shock factor 1 (HSF1) as a key player [22,23,24].

### 1.2. Heat Shock Proteins (HSPs) Are Multifamily Chaperones Implicated in Several Malignancies

HSPs have been traditionally grouped into six main families according to their molecular weight [25,26]. These include small HSPs (sHSPs), HSP40 (DNAJ), chaperonin or HSP60, HSP70, HSP90 and large HSPs (HSP110 and glucose-regulated protein 170, GRP170) [27]. Due to growing numbers of HSP members and their diverse and/or overlapping structures and functions, Kampinga et al. have set a new classification of HSP families which includes alphabet letters A, B, C, D, E, H rather than the traditional molecular weight system [27], see also Table 1.

High expression levels of HSPs have been reported in many cancers, including breast, head and neck, gallbladder, colorectal, skin, liver, colon, renal, prostate as well as ovarian cancer [21,28]. Of particular interest, HSPs play dual complex role in apoptosis via promoting or counteracting cell death. For instance, HSPs have been shown to activate apoptotic mediators such as pro-caspase 3 [29,30] and conversely, they bind and inhibit several molecules at different levels in the apoptotic pathway [31]. Among the anti-apoptotic events is the blockade of cytochrome C and SMAC Diablo release from the mitochondria by HSP27 besides antagonizing caspase 3 and 9 [32,33,34]. HSP27 can also suppress other apoptotic death receptor pathways, including TNFα, Fas and TRAIL [35]. Similarly, HSP70 inhibits apoptosis by interfering with the c-jun kinase pathway and preventing cytochrome C release from mitochondria [34,36] (See Figure 2). Moreover, HSPs have been found to chaperone several oncogenes including mutant P53 and prevent its degradation, thus evading the classical apoptotic pathway and resulting in cancer cell survival [37,38,39]. Furthermore, increased levels of certain HSPs conferred drug resistance in many cancers including prostate [40,41], liver [42], lung [43], colon [44], head and neck [45] and ovarian cancer [6].

Having anti-apoptotic properties and drug resistance characteristics, overexpression of HSPs in variant malignancies has been correlated to cell survival, tumor progression and metastasis as well as poor prognosis [47,48]. Therefore, most studies consider HSPs not only as diagnostic/prognostic markers but also as ideal therapeutic targets for cancer therapy [21,49,50]

By virtue of the increasing interest in HSPs as a potential drug target for cancer treatment among gynaecologists, we focused on the function of HSPs in ovarian cancer and highlighted their roles in carcinogenesis and therapeutic resistance. We will start by briefly discussing the general function of HSPs in ovaries in both physiological and pathological conditions.

## 2. Biological Functions of HSPs in Healthy and Diseased Ovaries

Previous in vitro and in vivo studies have demonstrated the importance of HSPs in the development of normal ovaries, growth of ovarian follicles and their resistance to stress conditions. In swine, thermal stress and serum deprivation induced high transcription levels of HSP70.2, HSP72 and HSP105/110 in both granulosa cells and ovarian follicles. Moreover, the expression levels of the respective HSPs was reduced following hormonal treatment highlighting the regulation of stress related changes by hormones in ovarian tissues [51]. In rat, treatment of immature granulosa cells with follicle stimulating hormone (FSH) resulted in cell rounding concurrent with activation of p38 mitogen activated protein kinase (MAPK) pathway and HSP27 phosphorylation [52].

Notably, chaperones including HSP90 and HSP70 play a key role in regulation of the function of steroid hormones by modulating their receptor activity such as estrogen receptor (ER), progesterone receptor (PR) and androgen receptor (AR) [40,53,54,55]. Expression of HSP70 has been also described in normal ovaries where a chaperone complex including HSP70 and HSP90 is suggested to bind steroid receptors and regulate their function [56]. For that, two hypotheses have been suggested to modulate steroid receptor function; the first hypothesizes the association of a chaperone heterocomplex including HSP90, HSP70, HSP40, p23, etc. to the unbound receptors and keep them in an inactive state [57]. Binding of the ligand to steroid receptor results in dissociation of the complex and release of HSP70 and HSP90 chaperones. Since proliferation of the growing follicles in proestrus occurs because of sex steroids, HSP70 has been thought as inhibitor of steroidal effects [56]. Additionally, elevated HSP70 levels have been shown to repress steroid biosynthesis and secretion [58,59]. Heat shocked rat luteal cells lost their ability to synthesize or secrete LH-sensitive progesterone and 20α-dihydroprogesterone after treatment with 8-bromo-cAMP and forskolin [58]. Conversely, the other hypothesis postulates that HSP90/HSP70 machinery is essential for maintaining the appropriate conformation required for hormone-binding activity of the receptor [60]. Therefore, it is collectively apparent that HSPs modulate ovarian physiology via controlling sex steroid receptors functionality as well as regulating apoptotic mechanisms [6,61,62].

On the other hand, HSPs have been associated with cystic ovarian disease (COD). Expression profiles of HSP27, HSP70, HSP60 and HSP90 revealed abundant levels of HSP27 in primary, secondary, tertiary and cystic follicles and diminished in atretic follicles [63]. Furthermore, overexpression of HSP70, HSP60 and HSP90 has been noticed in tertiary and atretic follicles. As a conclusion, the aberrant expression of HSPs in ovarian cysts is suggested to counteract apoptosis and delay regression of cystic follicles [63,64]. Interestingly, the herbicide atrazine, which dysregulates estrous cycle in rats and impairs folliculogenesis, has been shown to reduce expression of HSP90 and increase follicular atresia [65]. Additionally, in rats, ACTH or cold stress-induced polycystic ovary syndrome (PCOS) reveal a significant elevation of the expression of HSP90 and abnormal ovarian morphology compared to the control group [66]. Furthermore, proteomic studies in women with PCOS have demonstrated two-fold increase in HSP90B1 levels suggesting a role in promoting cell survival and suppression of apoptosis [67].

## 3. Heat Shock Factor 1 (HSF1) in Ovarian Cancer

As briefly introduced, the heat shock response (HSR) is a cytoprotective physiological response in all mammals to resist various stresses. Notably, the same response accompanies different pathological conditions as well as many cancers [68,69]. It is well established that HSF1 is a key mediator of this response which induces the expression of HSP or chaperone genes to enable the stressed cells recover from potential damage [22,70]. Accumulating evidence has suggested the contribution of HSF1 to tumorigenesis. For instance, HSF1 has been demonstrated to control several genes that promote the transformed phenotype such as those involved in signaling, metabolism, cell-cycle regulation, adhesion and translation [71,72]. Moreover, overexpression of HSF1 has been reported in a multitude of cancers including liver, lung, breast and colon cancers where high HSF1 levels were related to unfavorable prognosis [71,73]. Furthermore, HSF1 knock-out mice are refractory to chemically-induced tumors [22] and mouse embryonic fibroblasts lacking HSF1 are resistant to oncogene-induced transformation [22].

Powell et al. have studied the implication of HSF1 in epithelial-to-mesenchymal transition (EMT) and TGFβ signaling in the ovarian cancer cell lines SKOV3 and HEY that were knocked down for HSF1. Interestingly, the expression of fibronectin that is known to promote the EMT following induction by TGFβ was dramatically reduced either under basal or TGFβ-induced conditions [74] strongly supporting the implication of HSF1 in TGFβ signaling as well as EMT in OC [74].

### Targeting HSF1 in Ovarian Cancer

Since HSF1 has been shown to be overexpressed in OC tissues and HSF1 is involved in tumor development and metastasis, Chen et al. have investigated its targeting as a potential therapeutic strategy against human EOC [75]. HSF1 knock-down in SKOV3 using specific shRNA cells could downregulate HSF1, leading to marked antitumor consequences, including increased apoptosis and reduced proliferation. Moreover, an animal study carried out by the same group confirmed the tumorigenic tendency of HSF1 expressing cells as injection of HSF1-deficient cells into immunodeficient nude female mice displayed no tumorigenesis until 39 days post-injection whereas injection of the control cells formed obvious tumors after 14 days [75]. Consistent with these results, in vitro and in vivo studies have revealed that targeting HSF1 using the nucleoside analogue (Ly101-4B) yields potent anticancer activity in epithelial ovarian cancer [76].

## 4. HSPs in Ovarian Cancer

### 4.1. HSPC (HSP90) Family

Mammalian HSPs comprise four main HSP90 proteins that have molecular mass of about 90 kDa and resides in different cellular organelles, including the ER, the mitochondria and the cytosol [77]. HSP90 members are essentially involved in key regulatory and oncological pathways (Figure 3) [21].

It is apparent that all HSP90 members are involved in the pathogenesis of ovarian cancer. Overexpression of HSP90 has been reported in ovarian carcinoma where it was linked to the International Federation of Gynecology and Obstetrics (FIGO) stage of the disease [78,79]. Additionally, many reports denote the association of high HSP90 levels with tumor aggressiveness, metastasis and resistance to chemotherapeutics [41,48,80,81,82]. Advanced serological approaches have identified HSP90 among the tumor antigen proteins in OC [80]. mRNA and proteomic analysis of 17AAG treated OC cell lines, A2780, have revealed increased expression of HSP72, HSC70, HSP27, HSP47 and HSP90B1 at the mRNA level. At the protein level, expression levels of the heterochromatin protein 1 were increased while expression of the histone acetyltransferase 1 and the histone arginine methyltransferase PRMT5 was reduced. The observed changes following HSP90 inhibitor 17AAG treatment indicate a complex molecular roles of HSP90 in OC cells [83].

Tumor necrosis factor receptor-associated protein 1 (TRAP1), the mitochondrial homologue of HSP90, is significantly involved in several cancers including ovarian cancer. TRAP1 has been strongly expressed in tumor cells such as breast, colon, pancreas and lung whereas basal expression was detected in corresponding normal cells [84,85]. Interestingly, recent data from large scale studies demonstrated that lower TRAP1 levels that have been surprisingly observed in ovarian cancer are compatible with bad prognosis [85,86,87]. Moreover, TRAP1 expression has been found to correlate inversely to tumor grade or stage and correlate directly to the overall survival [87]. These results are in line with studies, in which a better response to chemotherapeutics in patients showing higher expression levels of TRAP1 was observed leading thus to the assumption that TRAP1 acts likely as a tumor suppressor [88]. Furthermore, Amoroso et al., 2016 suggested that the decrease in TRAP1 expression in ovarian cancer might be due to genetic deletion or gene-level copy number variations (CNVs) particularly in late stages of high-grade serous OC [89]. 

Of note, recent insights on ovarian cancer assign TRAP1 a key metabolic role in disease progression, platinum response and inflammatory activation [90]. TRAP1 can inhibit the mitochondrial respiratory chain through its direct interaction with the mitochondrial subunit B of SDH (SDHB). This significant effect confers survival to cancer cells and supports a mainly glycolytic type or Warburg phenotype of metabolism indicating that in certain types of cancers, TRAP1 can also be considered as pro-oncogene depending on the metabolic features of the tumor tissue [85,90]. In support to these intriguing observations on the diverse TRAP1 roles, it has been reported that TRAP1 constitutes a molecular complex with the cytosolic homologue HSP90 and cyclophilin D that serves to suppress apoptosis via regulating the mitochondrial transition pore opening [91].

Interestingly, the metabolic alteration effects of TRAP1 seemed to greatly influence the inflammatory response in terms of cytokines and chemokines. Upregulation of IL-6 and CSF2, two significant mediators of inflammatory response, has been demonstrated in TRAP1 deficient cells [90].

#### 4.1.1. HSP90 and Therapeutic Resistance

Certain HSP90 isoforms such as TRAP1 can modulate the responsiveness to anti-cancer drugs such as cisplatin in ovarian cancer [90]. Together with HSP27 and HSP70, the mRNA levels of TRAP1 were significantly up-regulated in cisplatin resistant OC cell lines compared to sensitive cells [92]. Interestingly, overexpression of TRAP1 has been considered an important factor in determining the degree of drug resistance in OC cells. PE01CDDP cells, which express relatively high TRAP1 levels, were far resistant, by twenty times, to cisplatin when compared to the lower cisplatin-resistant parental cells, PE01 [93,94]. In addition, other HSP90 proteins have been proven to confer resistance in OC cells as deduced following their targeting (see the next section).

#### 4.1.2. Targeting HSP90 in OC

The fact that HSP90 contributes to cancer progression and metastasis has rendered it an ideal molecule to target in several malignancies. In ovarian cancer, previous studies have shown that targeted inhibition of HSP90 is advantageous in terms of wide inhibition of numerous oncoproteins in EOC (see Figure 1 and Table 2). For instance, ganetespib as a monotherapy or in combination with paclitaxel showed marked reduction in cell growth, cell cycle arrest and induced apoptosis in vitro as well as ovarian tumors in transgenic mice in vivo [95]. Radicicol is another HSP90 inhibitor that has been shown to potentiate TRAIL mediated apoptosis in epithelial ovarian adenocarcinoma [96]. Moreover, since HSP90 is involved in the folding and stability of key mutant oncogenic proteins such as mutant p53 protein [97,98], targeting HSP90 disrupts HSP90/mutant p53 protein complex resulting in exposure of the mutant p53 to degradation by MDM2 and CHIP E3 ubiquitin ligases [98]. As an evidence, HSP90 inhibition exhibited strong cytotoxicity in p53 mutant cancer cells and xenografts [6,99]. Similarly, indirect targeting of multiple RTK receptors using HSP90 inhibitors, which interrupt the downstream pathways resulted in profound pro-apoptotic and anti-proliferative effects [100]. AUY922 that inhibits HSP90 has been tested alone or in combination treatment with carboplatin where it suppressed cell proliferation and significantly reduced carboplatin IC50 [101]. In line with the previous findings, recent studies have shown that the HSP90 inhibitor ganetespib potentiates the cytotoxic effect of carboplatin even in tumor cells lacking wild-type p53 [102]. Combined treatment of the two drugs led to persistent DNA damage and massive global chromosome fragmentation through inhibition of DNA repair and cell cycle control mechanisms [102].

#### 4.1.3. Diagnostic and Prognostic Value of HSP90 in OC

Serological screening in patients with OC has identified HSP90 among the tumor antigens. These included other molecules like S18, JK-recombination signal binding protein, CDC23, ribonucleoprotein H1, RAN binding protein 7, TG-interacting factor, eukaryotic translation initiation factor p40 subunit, ribosomal protein L8, human amyloid precursor protein-binding protein 1, IQ motif containing GTPase activating protein 1 and ribosomal protein L3 [80]. Further analysis of HSP90 autoantibodies prevalence revealed that HSP90 was linked to late stage of OC, suggesting its use as a potential biomarker [80]. Other previous studies have shown high expression of HSP90 and HSP70 where HSP90 was directly correlated to levels of sex steroid receptors in the tumor cells [105].

GRP94 or HSP90B1, the ER resident isoform of HSP90, has been recently reported among the plasma biomarkers associated with EOC [106]. In this study, other plasma biomarkers including IFNγ, IL-6, IL-8, IL-10, TNFα, placental growth factor (PlGF) were co-assessed with HSP90B1 and adjusted for the well-known cancer antigen CA-125 that is widely expressed in most ovarian cancers [107,108,109]. Interestingly, after adjustment for CA-125 and out of all measured biomarkers, HSP90 could significantly predict the presence of early EOC suggesting its usage as a disease-predictive biomarker [106]. In patients with recurrent advanced stage ovarian carcinoma the synergistic value of HSP90 inhibitors as co-therapy with platinol and paclitaxel was utilized, although the prognostic value of HSP90 in the effusions was not considered [110].

### 4.2. HSPA (HSP70) Family

The HSP70 family has been broadly studied in various stress and disease conditions. In human, there are 13 members within this family and they exhibit varying degrees of structural similarity and functionality [27,111]. Although most HSP70 members are widely localized in the cytoplasm and nucleus, certain members of HSP70 occupy distinctive cellular organelles like GRP78 or BiP (HSPA5) in the ER and GRP75 or mortalin (HSPA9) in the mitochondria [27]. 

Overexpression of HSP70 has been linked to the aggressiveness of ovarian cancer [112]. Moreover, supported by both cell culture and xenograft mouse model, HSP70-2 has been recently shown to support tumor growth and invasion in EOC via modulating several cellular events including cell cycle, apoptosis and epithelial mesenchymal transition pathways [113]. Furthermore, Koshiyama et al. revealed a strong correlation between HSP72 expression and p-53 positive ovarian tumors [62]. Studies on the effusions from OC patients revealed an association between HSP70 and poor overall survival [14,114]. Nevertheless, former studies lessened the significance of HSP70 in prognosis of epithelial ovarian carcinoma and correlated its expression with FIGO cancer stages [78].

Mortalin or HSPA9, the mitochondria-resident HSP70 isoform, has been also implicated in ovarian carcinogenesis and tumor malignancy [115]. Microarray results obtained from ovarian cancer tissue have revealed that mortalin is abundantly expressed in advanced stages compared with early stages of ovarian carcinomas and normal ovarian tissues [116]. Overexpression of mortalin and its capability to induce malignancy comes likely from its binding to the cytoplasmic P53 [117]. Hu et al. have demonstrated that mortalin displays its oncogenic role in ovarian cancer by promoting tumor growth and migration/invasion via crucial pathways including cell-cycle and the MAPK–ERK signaling pathways [118]. Other reports have shown that mortalin inhibition could suppress serous ovarian carcinoma cell proliferation, cell motility and EMT progression via inhibition of Wnt/β-Catenin signaling pathway [119]. Moreover, recent studies have demonstrated that expression of mortalin in OC cells is regulated via association of the NF-κB p65 to the mortalin promoter [120].

Similar to other HSP70 homologues, the ER localized chaperone GRP78 contributes to the OC development and progression [121]. Humoral response against GRP78 has been initially reported in OC patients in 1997 [122]. Interestingly, sera collected from ovarian cancer patients could detect GRP78 in cancerous ovarian tissues but not normal ovaries suggesting that the existence of GRP78 antigen is specific to OC [122,123]. Taylor et al. have shown that the expression of GRP78 can be used to discriminate between early stage and stage III/IV ovarian cancer [124]. Nevertheless, controversial results have demonstrated lack of difference between the levels of GRP78 autoantibodies in ovarian cancer and control patients [125]. Recent investigations have revealed overexpression of membrane GRP78 in OC and its positive correlation with proliferation [126]. With regard to these variations in the level of GRP78 autoantibodies in OC patients, Delie et al. have suggested that these differences might be due to variant methods used (ELISA or immunoblot) or unequal sample size [121].

#### 4.2.1. Therapeutic Resistance and Targeting of HSP70 in OC

It has been proven that HSP70 is highly expressed in resistant OC cells and its overexpression counteracts cisplatin-induced apoptosis by preventing Bax translocation to the mitochondria and subsequent mitochondrial protein release to cytosol (see Figure 2A for HSP70 cancer promoting functions) [127]. Overexpression of HSP70 has been detected in several OC cell lines and cells derived from patients following manumycin, a famesyl transferase inhibitor (FTI), treatment. Up-regulation of HSP70 has been suggested as a cytoprotective response and resistance strategy against FTIs induced apoptosis in cancer cells [128]. HSPA6, a cytosolic HSP70 member associated with heat stress, has been recently reported to resist Magnetic Fluid Hyperthermia MFH-based treatment of ovarian cancer. Inhibition of HSPA6 using siRNA or 2-phenylethyenesulfonamide (PES) led to enhanced OC cell death following exposure to MFH [129]. In addition, ovarian cancer cells which highly express GRP78 showed resistance to paclitaxel treatment. This refractory response was markedly altered upon targeting GRP78 using siRNA where the cells showed high sensitivity to paclitaxel [130]. As a support for these findings, Li et al. have reported weak staining of GRP78 in the chemotherapy-sensitive ovarian tumor sections compared to strong staining in the cisplatin-resistant C13K cells [131]. Moreover, the same researchers found that GRP78 knockdown in cisplatin-resistant OC cells could rescue the senescence sensitivity to cisplatin [131]. Collectively, these data highlight the contribution of HSP70 members to therapeutic resistance in OC and suggest the potential of their targeting for OC treatment as presented in Table 3.

Despite the promise of using HSP inhibitors in cancer treatment regimens either alone or in combination with other drugs, there exist certain limitations that should be considered. For instance, the high sequence homology among HSP members within the same family, which may reach 80–100% in case of HSPA family [111], may hamper specific HSP targeting and in many cases can produce cytotoxic effects. Another obstacle for using HSP inhibitors, in general, in cancer treatment is that silencing single HSP member may not be as efficient as proposed because of functional compensation by other HSP homologues [132]. This conclusion has recently been evidenced by observations reported by Prince et al., who demonstrated that dual targeting of HSP70 and HSP90 in bladder cancer cells is more advantageous than single HSP inhibition [133]. Furthermore, it has been reported that treatment of cancer cells with HSP inhibitors or proteasome inhibitors results in HSF1 activation and compensatory induction of HSPs thereby reducing the antitumor activity of such inhibitors [134]. Other limitations in preclinical studies testing HSP inhibitors are the usage of cell lines such as A2780, HeyA8, and SKOV3. Proteomic and genomic analyses have revealed that these cell lines and others are poor models for HGSC [135,136].

#### 4.2.2. Diagnostic and Prognostic Value of HSP70 in OC

The diagnostic value of HSP70 in ovarian cancer patients has been recently discussed by Kang et al. [137]. Expression analyses of Fas-associated factor 1 (FAF1) and heat shock protein 70 (HSP70) revealed lower expression of FAF1 while concurrent increase in HSP70 levels in ovarian cancer compared to their levels in normal ovary [137]. Moreover, both proteins were differently linked to the tumor stage. FAF1 showed reduced expression levels in advanced stages of OC, namely stage III and stage IV, compared with early stages (stage I or stage II). On the other hand, HSP70 was predominantly overexpressed in papillary serous carcinomas and undifferentiated ovarian cancer. Taken together, these data indicate a potential for the use of the characteristic FAF1/HSP70 inverse expression pattern to predict OC [137]. Furthermore, recent proteomic based studies have suggested HSP70 as a diagnostic yardstick in OC and revealed potent antibody response against HSP70 in OC sera compared to normal individuals [138]. Additionally, in vitro experiments as well as data from Oncomine database and Cancer Cell Line Encyclopedia (CCLE) have elucidated a strong association between mortalin expression and serous ovarian carcinoma suggesting its promise use as a diagnostic biomarker in serous ovarian carcinoma [119].

### 4.3. HSPD and HSPE (Chaperonin) Family

The well-known candidate in the HSPD family is HSPD1/HSP60 that is traditionally known as 60 kDa chaperonin and is mainly localized in the mitochondria [139,140]. It has been demonstrated that HSP60 regulates cell cycle and apoptosis in cancer via survivin and p53 mediated mechanism (Figure 2B) [46]. This mitochondrial chaperone has been also reported to exist in different cellular compartments including the cytosol and the nucleus as well as the extracellular environment [139]. Together with the 10 kDa co-chaperone, HSP60/HSP10 complex, it performs crucial cellular activities including protein folding, transport of proteins across membranes and other non-chaperone functions [14]. Therefore, HSP60 has been referred as moonlighting protein [141].

The fact that HSP60 has been widely involved in several malignancies evoked many researchers to investigate its potential role in ovarian cancer. Initial studies assessing HSP60 in OC patients revealed detectable yet variable mRNA levels of HSP60 in tissues of ovarian carcinoma [142]. Recent reports have demonstrated decreased overall survival in patients with advanced OC who express high HSP60 levels suggesting its use as a potential prognostic biomarker [143]. In line with the previous findings, Bodzek et al. have shown that immunoglobulins against HSP60 and HSP65 were correlated to the stage of the neoplastic process. For instance, the expression of HSP60 was remarkably higher at the early stages of OC then decreased with advanced stages [144].

In the context of cooperativity between HSP60 and HSP10 (HSPE1) in terms of forming functional molecular complex, HSP10 has been also shown to contribute in a certain way to OC. Akyol et al. have demonstrated that HSP10 can modulate the immune response in patients with advanced ovarian malignancies. Consistent with its detection in both sera and ascitic fluids of ovarian cancer patients, HSP10 has been found to suppress the expression of T cell receptor (TCR)-associated signal transducing zeta chain (CD3-zeta) leading to impaired immune responsiveness of T cells and ultimately tumor-mediated T-cell dysfunction [145]. Furthermore, DNA microarray technology has been exploited to identify differentially expressed genes in chemosensitive and chemoresistant ovarian serous papillary carcinomas in a study including 158 patients. Interestingly, gene expression analysis as well as immunohistochemistry have identified HSP10 as an independent factor of progression-free survival [146]. 

#### 4.3.1. HSP60 and Therapeutic Resistance

Early investigations have shown the implication of HSP60 in OC resistance to chemotherapeutics. Compared with controls, HSP60 transcripts were remarkably abundant in A2780 human ovarian carcinoma cells that were selected for cisplatin or oxaliplatin resistance. These uneven mRNA levels of HSP60 denote a strong association with in vitro resistance to platinum compounds [147]. In accordance with these results, it has been recently shown that targeting HSP60 sensitizes variant resistant OC cell lines to docetaxel or cisplatin treatment and results in significant cytotoxic effects [148]. A study by Kamishima et al. revealed the contribution of cytosolic HSP60 (c-HSP60) in conferring resistance to OC cell lines [149]. In this study, HSPs levels and roles were compared in two human ovarian cancer cell lines; the first, TYK-R10, which resists cisplatin and exhibits cross-resistance to anti-cancer drugs including adriamycin (ADR), vincristine (VCR) and etoposide and the second, is the parental line (TYK-nu). Under normal culture condition, the cisplatin-resistant TYK-R10 cells have been found to significantly express c-HSP27, c- HSP60, c-HSP70 and n-HSP70 compared to diminished levels in TYK-nu cells. Strikingly, while heat shock treatment augmented cisplatin resistance in TYK-R10, but not TYK-nu, the resistance of TYK-nu to ADR was significantly increased compared to TYK-R10. The acquired resistance to ADR has been attributed to overexpression and cellular relocalization of HSP27 and HSP60 in TYK-nu cells [149].

#### 4.3.2. Targeting HSP60 in OC

Though its involvement in resistance to anti-cancer drugs, therapeutic interventions that target HSP60 in OC is relatively low as compared to other HSPs. Nevertheless, we would say that anti-HSP60 drugs are emerging rapidly. Meng et al. have recently shed the light on HSP60 modulators, including inhibitors from both natural products and synthetic compounds, which are expected to be largely used in cancer research including OC in the near future [139]. 

### 4.4. DNAJ (HSP40) Family

DNAJ or HSP40 is the largest HSP family in humans comprising almost 50 members [27]. Structurally, DNAJ is characterized by a conserved J-domain that is essential for the recruitment and stimulation of the HSPA ATPase activity [150,151]. Despite its implication in many human malignancies such as lung [152], gastric [153], colorectal [154] and cervical cancers [155], members of this family are less studied in ovarian cancer.

Though scarcity of HSP40 studies on OC, an interesting observation by Shridhar et al. has led to the identification a gene that is in part homologous to the DNAJ domain, which exists in a several proteins including HSP40 family [156]. The identified gene has been designated as methylation-controlled J protein (MCJ). Strikingly, unlike normal ovarian cells, loss of MCJ expression has been reported in OC cell lines. Additionally, in patients with primary ovarian tumors, MCJ downregulation or complete absence of expression has been reported in 67% of the cases. Furthermore, treatment of OV202 cells with 5-Aza-2’ -deoxycytidine caused upregulation of MCJ in in a dose-dependent manner. Altogether, these results suggest that MCJ loss may serve as a potential prognostic factor in OC and may confer resistance to OC chemotherapeutics [156].

### 4.5. Small Heat Shock Proteins (sHSPs)

This class of molecular chaperones include proteins with molecular weight of 12-43 kDa [157]. sHSPs are present in different cellular locations including cytoplasm, nucleus and plasma membranes and serve primarily to prevent aggregation of denatured and misfolded proteins [158]. Members within this family are functionally distinguished from large HSPs because they lack ATPase activity and they have broad substrate specificity; therefore, they are commonly known as “holdases” [159]. In addition to their chaperone function, sHSPs are widely implicated in pivotal biological processes including cell proliferation, apoptosis, stress defense, cell cycle regulation and, of critical importance, cellular transformation to malignant phenotypes [35,160,161]. Notably, the function of key sHSPs is tightly regulated by post-translational modifications like phosphorylation (see Figure 4A for general HSPs’ structure including small HSPs). It has been shown that aberrant phosphorylation of sHSPs is correlated with cancer development and progression [162,163]. Additionally, sHSPs have been known for anti-apoptotic properties in cancer cells (Figure 2C) [34,164,165].

HSPB1 or HSP27 is an eminent candidate of sHSPs family which has been extensively studied in many cancers, including breast cancer [166], endometrial cancer [167], lung cancer [168], liver cancer [169] andprostate cancer [170]. In ovarian cancer, numerous data indicate the involvement of HSP27 in OC pathogenesis and therapeutic resistance reflecting its significance in predicting of the disease stage and prognosis [171,172]. For instance, Geisler et al. have demonstrated an inverse relationship between HS27 expression level and FIGO stage of OC and proposed HSP27 as independent prognostic indicator of survival in patients with epithelial ovarian carcinoma even after longer follow-up [173,174]. Increased expression level of HSP27 has been reported to be associated with ovarian tumor progression and aggressiveness [175,176]. Further investigations confirmed the presence of HSP27 as well as hsp27-cytochrome c complexes in cell free endo-cervical or posterior vaginal preparations from women with endometrial or ovarian cancer [177]. In line with the previous findings, Olejek et al. have shown that sera of women suffering from OC contained higher levels of HSP27 antibodies compared with healthy ones [178]. Interestingly, both mRNA and protein levels were found to be clearly elevated in EOC patients with peritoneal metastasis in comparison with those without peritoneal metastasis [179]. These results strongly link the overexpression HSP27 in EOC to the incidence of peritoneal metastasis and subsequent poor clinical outcome [179]. In accordance with former reports, using ELISA assays, Zhao et al. have shown that serum levels of HSP27 are significantly high in EOC patients compared to patients with benign ovarian tumors and the overall increase in HSP27 levels were exclusively detected in patients with peritoneal metastases [180]. Taken together, the previous data suggest that HSP27 can be used as potential biomarker as well as indicator of ovarian cancer and its metastatic status [180].

HSPB5 or αB-crystallin (CRYAB) is a stress inducible chaperone that was originally identified as a major lens protein in the eye [163,181,182]. In response to stress situations like radiation and peroxidation, CRYAB exerts its chaperoning activity via binding unfolded or disordered proteins, increasing their solubility and hampering their undesirable hydrophobic interactions, thus, preventing their aggregation. As a consequence, it inhibits induction of apoptosis and promotes cell survival [158,183]. On the other hand, the oncogenic properties of CRYAB have been clearly documented in many malignancies such as lung cancer [184], head and neck cancer [185], breast cancer [186,187] and colorectal cancer [188].

Many reports have linked CRYAB expression to OC progression and poor clinical outcome [165,189,190]. Earlier studies have demonstrated that lower CRYAB expression is associated with adverse patient survival [191]. Recently, however, immunohistochemical analysis of CRYAB and p53 has revealed that both proteins are highly expressed in ovarian cancer specimens and their co-expression can serve as independent prognostic factor of disease-free survival (DFS) and overall survival (OS) [190]. For instance, data reported by Tan et al. have shown that patients with increased co-expression of CRYAB and p53 have the worst prognosis among individuals with ovarian cancer [190]. Importantly, experiments performed on the human serous ovarian cancer derived cell lines, OV-MZ-6 and HEY, have revealed that overexpression of CRYAB could significantly inhibit TRAIL as well as cisplatin induced apoptosis [189].

#### Therapeutic Resistance and Targeting of sHSPs in OC

Several reports pointed to the contribution of sHSPs to chemoresistance in variant cancer types including OC [6,160,172,192,193]. Upregulation of HSP27 has been strongly correlated with limited responsiveness to platinum-based and topoisomerase-II-directed chemotherapy [194,195]. In ovarian cancer, previous clinical trials such as GOG111 and OV-10 have demonstrated that combined paclitaxel and cisplatin chemotherapy showed higher efficacy as compared to cyclophosphamide and cisplatin co-treatment, the standard regimen at that time [196,197]. The favorable therapeutic outcomes for paclitaxel/cisplatin combination have been attributed to the ability of paclitaxel to suppress the HSP27 as shown by Tanaka et al. [198]. Additionally, the same group demonstrated a potential role for HSP27 in tubulin regulation and arrangement specially in the G2/M phase giving another explanation for the desirable therapeutic responsiveness upon silencing HSP27 in OC cells [198]. In agreement with these previous findings, Pai et al. have recently shown that (2-Methoxy-5-[2-(3,4,5-trimethoxy-phenyl)-ethyl]-phenol) or shortly MT-4 is able to suppress both sensitive A2780 and multidrug-resistant NCI-ADR/res ovarian cancer cell lines via downregulation of HSP27 and minimizing its interaction with caspase-3 [193]. Surprisingly, however, the in vitro results reported by Stope et al. indicated a heterogeneous expression pattern of HSPB1 upon treatment with paclitaxel and carboplatin, despite their anti-proliferative effect, on selected OC cell lines [172]. Interestingly YangZheng XiaoJi (YZXJ), the traditional Chinese anti-cancer medicine, has been demonstrated to suppress the phosphorylation of HSP27 and silencing of HSP27 enhances the cytotoxic effects of YZXJ in OC cells [199]. A summary of these studies and their effect is presented in Table 4.

### 4.6. Clusterin

Clusterin (CLU) is a chaperone protein whose properties resemble sHSPs in many aspects including cytoprotective as well as oncogenic criteria [201,202,203,204,205]. This molecular chaperone is a highly glycosylated glycoprotein with molecular mass of 80 kDa and its structure comprises two polypeptide chains linked together by four to five disulfide bonds [202,206] (see Figure 4B for CLU structural architecture). It is of note that CLU stands as one of the major extracellular chaperones that has been extensively investigated in many cancer types [201,203]. Owing to its shared characteristics with HSPs, we sought in this section, and in the context of HSPs, to review the relevant research on CLU and its role OC.

On the whole, CLU is broadly involved in the carcinogenesis, progression, metastasis and therapeutic resistance of myriad cancers [160,207,208]. These include, liver [209], breast [210], lung [211], prostate [212] as well as ovarian cancer and other cancer types [207,213]. Interestingly, it has been postulated that different CLU isoforms play controversial roles inside the cell. For instance, in ovarian cancer cells, the nuclear form of clusterin (nCLU) has been found to delay cellular growth and promote apoptosis [213,214], while its secreted form (sCLU) exhibits anti-apoptotic potential and consequently accounts for the emergence of chemoresistant and aggressive phenotype [213,215,216]. Consistent with the aforementioned observations, high sCLU expression has been found in recurrent-resistant, paclitaxel resistant as well as Taxol-resistant tumors [217,218]. Notably, CLU has been demonstrated to physically bind paclitaxel hampering its interaction with microtubules, thus preventing apoptosis induction by paclitaxel [219]. Moreover, various in vitro studies have shown the implication of CLU in progression of OC. Upon silencing CLU using shRNA, Wei et al., have indicated increased sensitivity of silenced OC cell lines to chemotherapy [213]. In addition, other effects were observed including diminished cell proliferation, migration and invasion [213]. These results were further supported by Fu et al., who used a lentivirus-based approach to silence CLU in OC cells. These findings revealed clear reduction in proliferation, clonability, migration, invasion of the cell lines used [220]. Likewise, studies including siRNA or OGX-011, a second generation antisense oligodeoxynucleotide against CLU, have been shown to modulate the responsiveness or sensitivity of OC cell lines to paclitaxel [218].

#### 4.6.1. CLU as a Prognostic Biomarker in OC

It has been reported that overexpression of CLU in OC is correlated with increased tumorigenesis, poor survival and unfavorable therapeutic outcome [218,219,221,222,223]. In a recent proteomic study by Zhang et al., CLU has been identified among specific upregulated proteins following surgical intervention in OC patients, suggesting its contribution to postoperative recurrence of epithelial ovarian cancer [224]. Moreover, in the same study high plasma levels of CLU have been detected in chemotherapy-resistant patients compared to chemotherapy-sensitive group. Taken together, these data strongly suggest CLU not only as a biomarker for OC prognosis but also as predictor of chemotherapy resistance in ovarian cancer [224].

#### 4.6.2. Targeting CLU in OC

Custirsen (OGX-011) is the most well-known anti-CLU drug in the oncology field. Despite being used intensely in preclinical and clinical studies in various cancer types, such as prostate [225], HCC [205] and lung cancer [226], very rare have studies incorporated custirsen in clinical trials regarding OC. For instance, OGX-011 has been used with docetaxel in phase I trial in patients with ovarian cancers together with others suffering from castration-resistant prostate cancer (CRPC), non-small cell lung cancer (NSCLC), breast, bladder and renal cancers. The outcomes suggested further inclusion of OGX-011 in combination with chemotherapy as a compound with demonstrable biological activity [227].

## 5. Conclusions and Perspectives

HSPs and CLU have been proven to play a key role in tumorigenesis and can be employed as potential biomarkers for clinical diagnosis and prognosis in patients with ovarian cancer. The majority of HSPs show increased expression in ovarian cancer tissues, where they share many carcinogenic actions, such as impediment of apoptosis and conferring drug resistance. These criteria have raised the possibility to target HSPs in order to treat ovarian cancer. In this area, most anticancer drugs have been designed to target HSP90 and HSP70. However, increasing research is currently directed to target other chaperones like HSP27 and CLU with promising results. Additionally, combination therapy by targeting two or more HSPs in cancer treatment regimens is in progress. On the other hand, there are many challenges in clinical development of HSP blockade due to undesired toxicity or limited efficacy in clinical trials. Therefore, new strategies to develop novel powerful and selective HSP inhibitors are essential reduce the burden of cancer in the upcoming future.

## Figures and Tables

**Figure 1 cancers-11-01389-f001:**
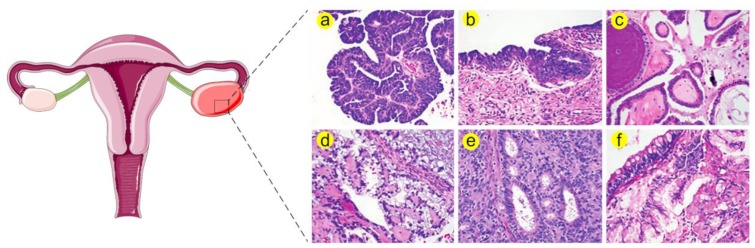
Histological stratification of ovarian cancer ^a^. (**a**) High grade serous carcinoma (HGSC) is distinguished by increased nuclear atypia, high nuclear-to-cytoplasmic ratio and abundant mitosis. (**b**) Serous tubal intraepithelial carcinoma (STIC) resembles HGSC in many morphological aspects such as severe atypia, defective cellular polarity and mitoses. Therefore, STIC is believed to be a precursor of HGSC. (**c**) Low grade serous carcinoma (LGSC) is characterized by increased papillae, mild nuclear atypia and low nuclear-to-cytoplasmic ratio. (**d**) Clear cell carcinoma exhibits large tumor cell sizes and frequent clearing of the cytoplasm together with stromal hyalinization. (**e**) Endometrioid adenocarcinoma can be differentiated by gland formation that recapitulates endometrial glands. This type is further categorized according to cellular architecture and nuclear atypia. (**f**) Mucinous adenocarcinoma is characterized by increased cellular mucin and formation of goblet cells. ^a^ Histological images are adapted from Nature Reviews Disease Primers [3].

**Figure 2 cancers-11-01389-f002:**
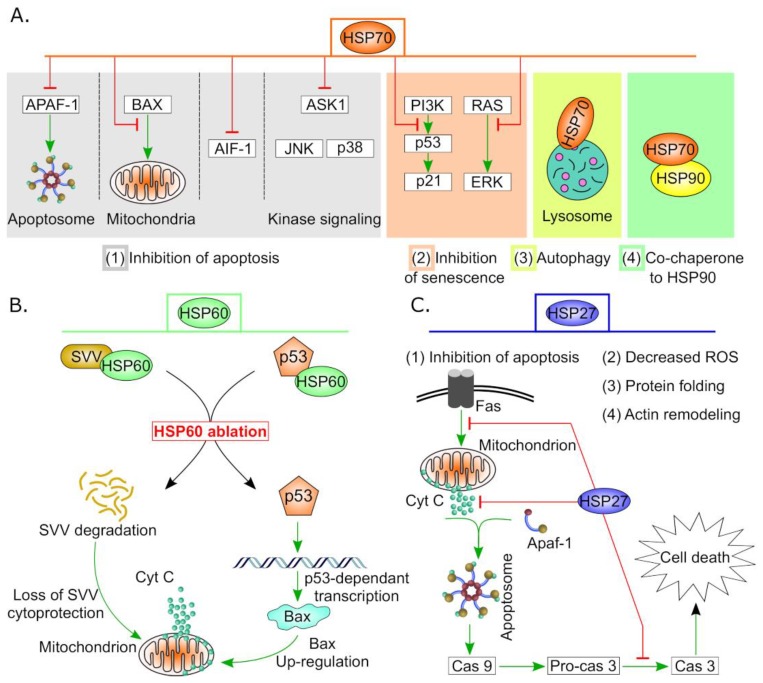
Anti-apoptotic and cell survival activities of some HSPs in cancer. (**A**) Variant roles of HSP70 in carcinogenesis. High expression of HSP70 in tumor cells suppresses apoptosis by (1) hindering APAF1 recruitment to apoptosome, interfering with BAX translocation to mitochondria, downregulation of AIF1 and other stress-related kinases. Additionally, (2) HSP70 regulates both p53-dependent and -independent senescence pathways, (3) supports autophagy by stabilization of lysosomal membrane and finally (4) it forms complex with HSP90 which is essential for efficient functionality. (**B**) HSP60 controls apoptosis by stabilizing mitochondrial survivin (SVV) and hindering P53 pro-apoptotic actions. HSP60 ablation results in degradation of SVV and activation of the mitochondrial apoptotic pathway. In addition, silencing of HSP60 increases P53 stability and subsequently, triggers p53-dependent transcription of apoptotic proteins such as BAX which promotes cell death [46]. (**C**) HSP27 performs multiple functions in cancer including protein folding, actin remodeling, minimizing oxidative stress and inhibition of apoptosis. Sample anti-apoptotic events of HSP27 are indicated by red blunt arrows.

**Figure 3 cancers-11-01389-f003:**
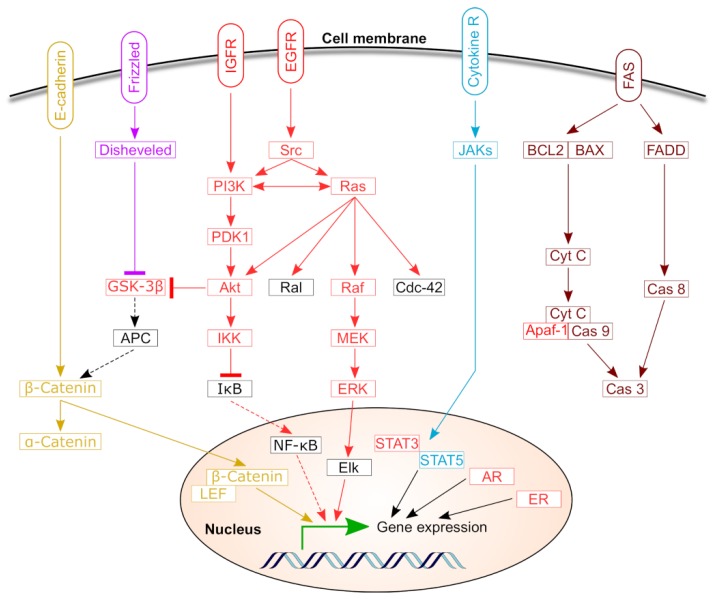
HSP90 functions as a mediator of many oncogenic and signaling pathways [21]. Several oncogenic proteins (shown in red) require HSP90 chaperonage for their proper folding and stabilization. For instance, HSP90 impacts the activity and/or function of receptor tyrosine kinases, serine-threonine kinases, steroid receptors, Src family members, telomerases and cell cycle proteins. Other distinctive pathways regulated by HSP90 are illustrated in different colors, including apoptotic pathway (brown), JAK/STAT pathway and cell-adhesion and Wnt-signaling (purple and light brown).

**Figure 4 cancers-11-01389-f004:**
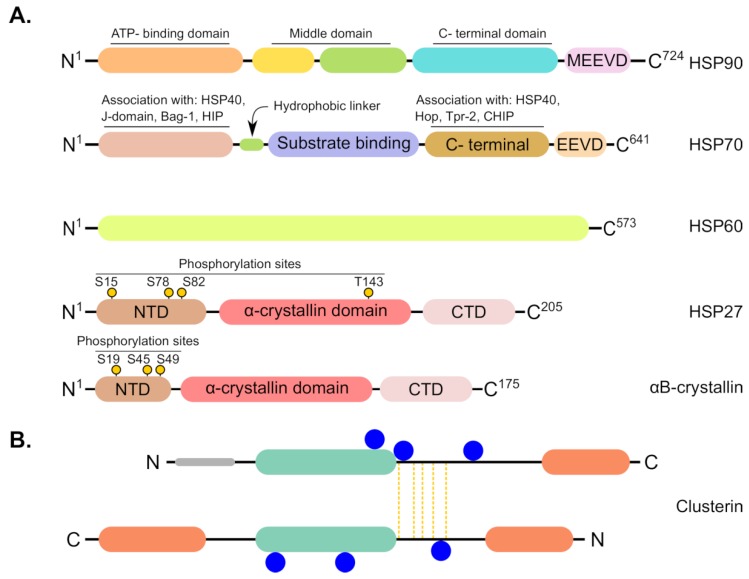
Schematic representation showing structural features of common chaperones involved in ovarian cancer. (**A**) From top to bottom; HSP90, HSP70 and HSP60 are ATP-dependent chaperones that harbor ATP binding sites within their structures whereas sHSPs such as HSP27 and αB-crystallin do not possess ATP binding sites. All HSPs have N-terminal and C-terminal domains (NTD and CTD) besides middle domain. sHSPs contain phosphorylation sites at specific serine (S) or threonine (T) residues, depicted as black sticks with yellow circles at their ends, and they are characterized by conserved α-crystallin domain that is flanked by variable N- and C-terminal ends. (**B**) The main structural topology of clusterin (CLU). Synthesis of CLU includes removal of the short 22-residue signal peptide (grey) as it enters to the ER lumen. Subsequent posttranslational proteolysis occurs in the Golgi where the protein is cleaved into α-(upper) and β-(lower) chains. The α- and β-chains are covalently connected by five disulfide bridges extending from ‘core region’ of both chains (yellow vertical lines). In addition, the α-chain is predicted to contain one amphipathic α-helix while the β-chain contains two α-helices. Moreover, both chains have a coiled coil structure (light green) and the mature protein is known to have six N-linked glycosylation sites (blue circles).

**Table 1 cancers-11-01389-t001:** General overview of human heat shock proteins (HSP) families and common members [27].

HSP Family	Recent Name	Number of Members	Common Members and Their Alternative Names
HSP110	HSPH	4	HSPH1 (HSP105)
			HSPH2 (HSP110, HSPA4 and APG-2)
			HSPH4 (HYOU1/Grp170, ORP150 and HSP12A)
HSP90	HSPC	5	HSPC2 (HSP90α, HSP90AA2, HSPCA and HSPCAL3)
			HSPC3 (HSP90β, HSP90AB1, HSPC2, HSPCB, D6S182, HSP90B, FLJ26984)
			HSPC4 (GRP94, HSP90B1, GP96, ECGP, TRA1, endoplasmin)
			HSPC5 (TRAP1, HSP75, HSP90L)
HSP70	HSPA	13	HSPA1A (HSP70-1, HSP72 and HSPA1)
			HSPA1B (HSP70-2)
			HSPA5 (BIP, GRP78 and MIF2)
			HSPA6 (Heat shock 70kD protein 6 and HSP70B′)
			HSPA8 (HSC70, HSC71, HSP71 and HSP73)
			HSPA9 (GRP75, HSPA9B, MOT, MOT2, PBP74 and mot-2)
Chaperonins	HSPD and HSPE	14	HSPD1 (HSP60 and GroEL)
			HSPE1 (HSP10, chaperonin 10 and GroES)
HSP40	DNAJ	50	DNAJA1 (DJ-2, DjA1, HDJ2, HSDJ, HSJ2, HSPF4 and hDJ-2)
			DNAJB1 (HSPF1 and HSP40)
			DNAJC1 (MTJ1, ERdj1, ERj1p and Dnajl1)
sHSPs	HSPB	11	HSPB1 (HSP27, HSP28, HSP25, HS.76067, DKFZp586P1322, CMT2F and HMN2B)
			HSPB4 (CRYAA, crystallin alpha A and CRYA1)
			HSPB5 (CRYAB, crystallin alpha B and CRYA2)

**Table 2 cancers-11-01389-t002:** Summary of studies and clinical trials related to HSP90 targeting in ovarian cancer (OC).

Compound Used for HSP90 Targeting	Effect/Mechanism	Reference
Ganetespib (small molecule inhibitor of HSP90)	-Cell-cycle arrest and induction of apoptosis in vitro. -Delayed growth of orthotopic xenografts and spontaneous ovarian tumors in transgenic mice. -Downregulation and suppression of numerous proteins associated with EOC progression	[95]
Ganetespib + paclitaxel	-Paclitaxel potentiated the activity of ganetespib both in cultured cells and tumors.	[95]
Ganetespib + siRNAs	-Synergistic effect	[95]
Radicicol	-Increased enhanced TRAIL-induced apoptosis-related protein activation, nuclear damage and apoptosis	[96]
Suberoylanilide hydroxamic acid (SAHA)	-SAHA is histone deacetylase inhibitor (HDACi) which targets the HSP90/mutant p53 protein complex and liberates mutP35 from the complex leading to its degradation	[97,98]
17-AAG or 17AAG + tyrosine kinase inhibitors	-Marked apoptotic effect was observed in SKOV3, OVCA429 and ES2 cells after using of 17-AAG alone or in combination compared to single tyrosine kinase inhibitor	[100]
AUY922 or AUY922 + carboplatin	-The HSP90 inhibitor AUY922 suppressed proliferation of OC cells and decreased carboplatin IC50	[101]
Ganetespib + carboplatin	-Marked synergistic action in terms of cytotoxicity in ovarian tumor cells lacking wild-type p53	[102]
Ganetespib + other anticancer drugs including niraparib, carboplatin, paclitaxel, gemcitabine (ongoing phase II trial)	-This study is known as European Trial on Enhanced DNA Repair Inhibition in Ovarian Cancer (EUDARIO) -It includes women with variant stages of ovarian cancer, fallopian Tube Cancer and primary Peritoneal Carcinoma -The trial started in 30 November 2018 and completion date are expected to be in 30 June 2022	ClinicalTrials.gov Identifier: NCT03783949
Ganetespib + paclitaxel (GANNET53, completed phase I and phase II trials)	-The addition of ganetespib, HSP90 inhibitor besides weekly paclitaxel did not improve survival in platinum-resistant epithelial ovarian cancer (PROC) patients	ClinicalTrials.gov Identifier: NCT02012192 [103,104]
AT13387 + talazoparib (phase I)	-AT13387 is an HSP90 Inhibitor, while talazoparib is a PARP inhibitor	ClinicalTrials.gov Identifier: NCT02627430
Onalespib (AT13387) + olaparib (ongoing phase I trial)	-AT13387 is an HSP90 Inhibitor, whereas olaparib is a PARP inhibitor -The trial started in 19 May 2017 and completion date is expected to be in 1 June 2020	ClinicalTrials.gov Identifier: NCT02898207

**Table 3 cancers-11-01389-t003:** Summary of pre-clinical studies targeting HSP70 members in OC.

HSP 70 Member	Targeting Approach	Effect	Used Cell Line/Model	Reference
HSPA5 (GRP78)	siRNA + paclitaxel	Marked reduction in cell viability	HO-8910	[130]
	Knocking down	Rescues senescence sensitivity to cisplatin through P21 and CDC2	C13K cells	[131]
HSPA6 (HSP70B’)	siRNA and 2-phenylethyenesulfonamide (PES)	Reduction of cell viability following exposure to magnetic fluid hyperthermia (MFH)	A2780 cp20 and HeyA8	[129]
HSPA9 (GRP75 or Mortalin)	shRNA + cisplatin	Decreased cell proliferation, potentiation of cisplatin-induced apoptosis and lowering cell invasion potential	A2780 and A2780 cisplatin resistant cells	[116]

**Table 4 cancers-11-01389-t004:** Summary of pre-clinical and clinical studies targeting HSP27 in OC.

sHSP	Targeting Compound	Effect or Mechanism	Used Cell Line/Model	Reference
HSP27	Paclitaxel	Suppression of HSP27 expression concomitant with cell growth inhibition	BG-1 ovarian cancer cells and HeLa uterine cancer cells	[198]
	Apatorsen (OGX-427)	The OGX-427, antisense inhibitor targeting HSP 27, caused marked reduction of CA-125 in a dose dependent manner	Phase I trial (OC patients)	[200]
	YangZheng XiaoJi (traditional Chinese herbal medicine)	Increasing sensitivity of cancer cells to chemotherapeutics via modulating phospho-HSP27 levels	A2780 and A2780-CP70, SKOV3 and COV504	[199]
	MT-4	Inhibition of tubulin polymerization and induction of apoptosis via hindering HSP27/caspase 3 interaction	A2780 and multidrug- resistant NCI-ADR/res human OC cell lines	[193]
CRYAB	None (CRYAB effect was proofed in vitro via overexpression)	Resistance of TRAIL- and cisplatin-induced apoptosis	OV-MZ-6 and HEY cells as well as tumor tissues from patients with OC	[189]

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
