# Peer review of "Heat Shock Proteins and Ovarian Cancer: Important Roles and Therapeutic Opportunities"

_cancers, 2019, doi:10.3390/cancers11091389_

Round 1
Reviewer 1 Report
In abstract, description of ovarian cancer and HSPs is too general to get an insight from this section. lt will be better to focus on CLU (clusterin). ln respect with OC, CLU in this section should be described in detail.
And also an official name, CLU should be used instead of 'clusterin' in the text.
Table 1. Contents of " Common members and their alternative names " is better lined rather than centered.
Table 2. Contents of "Effect/mechanism" is better lined rather than centered.
Figure 2. Numbering should be clearly described in detail (1, 2, 3, 4).
Figure 4. ls necessary a dashed line between figure4A and 4B?
ln line 490, a sentence (Zhao et al., 2014 ~) should be corrected.
Author Response
Response to the reviewers
We thank the reviewers for their suggestions and constructive comments, all of which have been addressed in the revised version of the manuscript. All changes made are marked.
Reviewer 1:
In abstract, description of ovarian cancer and HSPs is too general to get an insight from this section. It will be better to focus on CLU (clusterin). In respect with OC, CLU in this section should be described in detail.
And also an official name, CLU should be used instead of 'clusterin' in the text.
Response:
The abstract has been appropriately modified and highlighted the implication of CLU in OC is highlighted. We also used CLU throughout the manuscript as suggested.
Table 1. Contents of “Common members and their alternative names" is better lined rather than centered.
Table 2. Contents of "Effect/mechanism" is better lined rather than centered.
Response:
We followed the journal guidelines for text and table formatting. Therefore, we suggest to leave this part for further processing by the journal editing office.
Figure 2. Numbering should be clearly described in detail (1, 2, 3, 4).
Figure 4. ls necessary a dashed line between figure4A and 4B?
Response:
In the revised version of the manuscript, we clearly described the numbered items in Figure 2 as the reviewer suggested. In addition, we removed the dashed line between Figures 4A and 4B.
ln line 490, a sentence (Zhao et al., 2014 ~) should be corrected
Response:
- The sentence has been corrected in the revised version of the manuscript.

Reviewer 2 Report
The paper contains wide review of the current literature concerning the role of HSPs in (ovarian) cancer cell. The work is interesting, but I think that conclusions are too far reaching. In my opinion, actual therapeutic opportunities should be discussed with greater caution. The work would have greater value if the Authors sensitize scientific community to the difficulties in the field rather than show excessive optimism.
Targeting HSPs as an anticancer strategy theoretically looks promising but there are many obstacles. HSPs are highly evolutionarily conserved. E.g. main members of HSPA family share between 80-100% sequence similarity (HSPA1A, HSPA1B, HSPA1L, HSPA2, HSPA8). This causes many difficulties, from production of specific antibodies, to specific inhibition of protein function. Many papers describing evaluation of HSPA protein level in cancer tissues use unspecific antibodies and their conclusions are biased, as we do not know which HSPA protein has been detected. An example is here: doi: 10.1186/1477-7819-11-141 Authors used anti-HSPA2 antibody (Santa Cruz Biotechnology). In our experience this antibody is unspecific and detects also inducible HSPA1 and HSPA2 proteins and cognate HSPA8. In our opinion anti-HSPA2 Ab (Abcam, EPR4596) is the one which specifically detects HSPA2 (the results are in line with those obtained with our custom made, highly specific Ab).
It is also known, that silencing one of one HSP family member is not effective, as its function can be replaced by other homologue. Example work is here: doi:10.1371/annotation/5a7961d9-a7ea-4b10-9b48-5b106c405b02 (From the abstract: “Since several paralogs of Hsp70 proteins exist in cytosol, endoplasmic reticulum and mitochondria, we investigated which isoform needs to be down-regulated for reducing viability of cancer cells. […] We found that for significant reduction of viability of cancer cells simultaneous knockdown of heat-inducible Hsp70 (HSPA1) and constitutive Hsc70 (HSPA8) is necessary.)
Moreover, the treatment of cancer cells with HSP or proteasome inhibitors results in the HSF1 activation and compensatory induction of HSPs, therefore reducing the antitumor activity of such inhibitors reviewed in doi: 10.2174/1568009614666140122155942
I would like Authors to discuss these issues.
Minor points:
Table 1 shows incomplete list of HSP family members, thus I wonder if it is meaningful in this shape?
Line 174 – “control group” would be better than “normal group”
Line 184 “promote” not “promotes”
Line 202 “immunodeficient nude female mice” not “immunodeficiency nude female mice”
Line 208, 255 – I would not use “isoforms”, rather “proteins”
Table 2 – why do not mention clinical trials? (e.g. NCT03783949 European Trial on Enhanced DNA Repair Inhibition in Ovarian Cancer (EUDARIO), NCT02012192 GANNET53, NCT02627430 Talazoparib and HSP90 Inhibitor AT13387 in Treating Patients With Metastatic Advanced Solid Tumor or Recurrent Ovarian, Fallopian Tube, Primary Peritoneal, or Triple Negative Breast Cancer, NCT02898207)
Table 2, Table 4 – please provide the NTC number of clinical trial mentioned
Conclusions: please mention constraints concerned with potential anti-HSP therapies
Author Response
Response to the reviewers
We thank the reviewers for their suggestions and constructive comments, all of which have been addressed in the revised version of the manuscript. All changes made are marked.
Reviewer 2:
The paper contains wide review of the current literature concerning the role of HSPs in (ovarian) cancer cell. The work is interesting, but I think that conclusions are too far reaching. In my opinion, actual therapeutic opportunities should be discussed with greater caution. The work would have greater value if the Authors sensitize scientific community to the difficulties in the field rather than show excessive optimism.
Targeting HSPs as an anticancer strategy theoretically looks promising but there are many obstacles. HSPs are highly evolutionarily conserved. E.g. main members of HSPA family share between 80-100% sequence similarity (HSPA1A, HSPA1B, HSPA1L, HSPA2, HSPA8). This causes many difficulties, from production of specific antibodies, to specific inhibition of protein function.
Many papers describing evaluation of HSPA protein level in cancer tissues use unspecific antibodies and their conclusions are biased, as we do not know which HSPA protein has been detected. An example is here: doi: 10.1186/1477-7819-11-141 Authors used anti-HSPA2 antibody (Santa Cruz Biotechnology). In our experience this antibody is unspecific and detects also inducible HSPA1 and HSPA2 proteins and cognate HSPA8. In our opinion anti-HSPA2 Ab (Abcam, EPR4596) is the one which specifically detects HSPA2 (the results are in line with those obtained with our custom made, highly specific Ab).
It is also known, that silencing one of one HSP family member is not effective, as its function can be replaced by other homologue. Example work is here: doi:10.1371/annotation/5a7961d9-a7ea-4b10-9b48-5b106c405b02 (From the abstract: “Since several paralogs of Hsp70 proteins exist in cytosol, endoplasmic reticulum and mitochondria, we investigated which isoform needs to be down-regulated for reducing viability of cancer cells. […] We found that for significant reduction of viability of cancer cells simultaneous knockdown of heat-inducible Hsp70 (HSPA1) and constitutive Hsc70 (HSPA8) is necessary.)
Moreover, the treatment of cancer cells with HSP or proteasome inhibitors results in the HSF1 activation and compensatory induction of HSPs, therefore reducing the antitumor activity of such inhibitors reviewed in doi: 10.2174/1568009614666140122155942
I would like Authors to discuss these issues.
Response:
In the revised version of the manuscript, we included the relevant information suggested by the reviewer (changes are tracked).
Minor points:
Table 1 shows incomplete list of HSP family members, thus I wonder if it is meaningful in this shape?
Response:
We presented the HSP families and important members of HSPs which, in our view, are sufficient to provide the reader with general information about HSPs classification and important HSPs in OC. Detailed information on HSPs classification and their variant members have been previously discussed in many occasions and also included in our references’ list such as (Jee, 2016; Kampinga et al., 2009).
References:
Jee, H. (2016). Size dependent classification of heat shock proteins: a mini-review. J. Exerc. Rehabil. 12, 255–9. doi:10.12965/jer.1632642.321.
Kampinga, H. H., Hageman, J., Vos, M. J., Kubota, H., Tanguay, R. M., Bruford, E. a, et al. (2009). Guidelines for the nomenclature of the human heat shock proteins. Cell Stress Chaperones 14, 105–11. doi:10.1007/s12192-008-0068-7.
Line 174 – “control group” would be better than “normal group”
Line 184 “promote” not “promotes”
Line 202 “immunodeficient nude female mice” not “immunodeficiency nude female mice”
Line 208, 255 – I would not use “isoforms”, rather “proteins”
Response:
The corrections have been made.
Table 2 – why do not mention clinical trials? (e.g. NCT03783949 European Trial on Enhanced DNA Repair Inhibition in Ovarian Cancer (EUDARIO), NCT02012192 GANNET53, NCT02627430 Talazoparib and HSP90 Inhibitor AT13387 in Treating Patients With Metastatic Advanced Solid Tumor or Recurrent Ovarian, Fallopian Tube, Primary Peritoneal, or Triple Negative Breast Cancer, NCT02898207)
Table 2, Table 4 – please provide the NTC number of clinical trial mentioned
Response:
The revised version of the manuscript contains now the indicated clinical trials in Table 2 and we provide also the available NTC numbers and references of the clinical trials mentioned in Tables 2 and 4.
Conclusions: please mention constraints concerned with potential anti-HSP therapies.
Response:
We referred to this issue in the conclusion section of the revised manuscript as suggested by the reviewer.

Reviewer 3 Report
The authors present a detailed and comprehensive outlook on the therapeutic targeting of heat shock proteins for ovarian cancer therapy. Therapies for patients with platinum resistant disease are critically needed and targeting HSPs may be provide a good opportunity. The manuscript should be accepted following minor revisions.
Comments:
Text editing for English grammar is required - particularly in the abstract. The use of PARP inhibitors is mentioned. It should be noted that PARP inhibitors have mostly been successful and are approved for patients with platinum sensitive disease, not resistant disease. The manuscript makes reference to studies that focused on the use of the cell lines A2780, HeyA8, and SKOV3. These cell lines have been found to be poor models for HGSC (Domcke, Nat Comm, 2013 and Coscia, Nat Comm, 2016). This should be noted in the text. The sentence on line 388-389 does not make sense - "Initial studies revealed overall yet variable mRNA..." There is only 1 reference to Figure 4 in the text in regards to sHSPs. sHSPs should be designated in the figure. Clinical trials testing a clusterin inhibitor are mentioned. It would be useful to include a summary of the trial results. Reference numbers are duplicated in the references section.Author Response
Response to the reviewers
We thank the reviewers for their suggestions and constructive comments, all of which have been addressed in the revised version of the manuscript. All changes made are marked.
Reviewer 3:
The authors present a detailed and comprehensive outlook on the therapeutic targeting of heat shock proteins for ovarian cancer therapy. Therapies for patients with platinum resistant disease are critically needed and targeting HSPs may be provide a good opportunity. The manuscript should be accepted following minor revisions.
Comments:
Text editing for English grammar is required - particularly in the abstract. The use of PARP inhibitors is mentioned. It should be noted that PARP inhibitors have mostly been successful and are approved for patients with platinum sensitive disease, not resistant disease.
Response:
In the revised version of the manuscript, we edited the abstract and referred to the comments raised by the reviewer regarding the use of PARP inhibitors.
The manuscript makes reference to studies that focused on the use of the cell lines A2780, HeyA8, and SKOV3. These cell lines have been found to be poor models for HGSC (Domcke, Nat Comm, 2013 and Coscia, Nat Comm, 2016). This should be noted in the text.
Response:
In the revised version of the manuscript, we included the relevant information regarding the cell lines studied and added the respective references.
The sentence on line 388-389 does not make sense - "Initial studies revealed overall yet variable mRNA..."
Response:
In the revised version of the manuscript, we changed the sentence to read “Initial studies assessing HSP60 in OC patients revealed detectable yet variable mRNA levels of HSP60 in tissues of ovarian carcinoma cell lines” instead of “Initial studies revealed overall yet variable mRNA levels of HSP60 in tissues of ovarian carcinoma cell lines”.
There is only 1 reference to Figure 4 in the text in regards to sHSPs. sHSPs should be designated in the figure.
Response:
The general structure of HSPs including HSP90, HSP70 and HSP60 besides the structure of sHSPs including HSP27 and αB-crystallin have been presented in Figure 4. We referred to this issue in the text. Furthermore, we provided a brief description for all HSP members in the figure legend.
Clinical trials testing a clusterin inhibitor are mentioned. It would be useful to include a summary of the trial results.
Response:
In the revised version of the manuscript, we explained the scarcity of clinical trials concerning CLU inhibitors in OC. Most of OGX-011 clinical trials have been performed in patients with prostate and lung cancers as we mentioned in the manuscript. In addition, we previously reported these CLU relative trials also in our review on prostate cancer (Hoter et al., 2019). However, in the present review, we briefly referred to the phase I trial that included OC patients.
Reference:
Hoter, A., Rizk, S., and Naim, H. Y. (2019). The Multiple Roles and Therapeutic Potential of Molecular Chaperones in Prostate Cancer. Cancers (Basel). 11, 1194. doi:10.3390/cancers11081194.
Reference numbers are duplicated in the references section.
Response:
We updated the reference list and corrected the unintentional errors.
